# Novel Virulent Bacteriophage ΦSG005, Which Infects *Streptococcus gordonii*, Forms a Distinct Clade among Streptococcus Viruses

**DOI:** 10.3390/v13101964

**Published:** 2021-09-29

**Authors:** Jumpei Fujiki, Shin-ichi Yoshida, Tomohiro Nakamura, Keisuke Nakamura, Yurika Amano, Keita Nishida, Keitaro Nishi, Michihito Sasaki, Tomohito Iwasaki, Hirofumi Sawa, Hitoshi Komatsuzawa, Hiroshi Hijioka, Hidetomo Iwano

**Affiliations:** 1Laboratory of Veterinary Biochemistry, Department of Veterinary Medicine, Rakuno Gakuen University, Ebetsu 069-8501, Japan; j-fujiki@rakuno.ac.jp (J.F.); Shinichi.Yoshida@kaneka.co.jp (S.-i.Y.); s21941005@stu.rakuno.ac.jp (T.N.); s21661158@stu.rakuno.ac.jp (K.N.); s21661157@stu.rakuno.ac.jp (Y.A.); nk.0418trbr@gmail.com (K.N.); 2Department of Maxillofacial Diagnostic and Surgical Science, Field of Oral and Maxillofacial Rehabilitation, Graduate School of Medical and Dental Sciences, Kagoshima University, Kagoshima 890-8544, Japan; nsktr.248@gmail.com (K.N.); zio@dent.kagoshima-u.ac.jp (H.H.); 3Division of Molecular Pathobiology, International Institute for Zoonosis Control, Hokkaido University, Sapporo 001-0020, Japan; m-sasaki@czc.hokudai.ac.jp (M.S.); h-sawa@czc.hokudai.ac.jp (H.S.); 4Department of Food Science and Human Wellness, College of Agriculture, Food and Environment Science, Rakuno Gakuen University, Ebetsu 069-8501, Japan; iwasaki@rakuno.ac.jp; 5International Collaboration Unit, International Institute for Zoonosis Control, Hokkaido University, Sapporo 001-0020, Japan; 6One Health Research Center, Hokkaido University, Sapporo 001-0020, Japan; 7Department of Bacteriology, Graduate School of Biomedical and Health Sciences, Hiroshima University, Hiroshima 734-8553, Japan; komatsuz@hiroshima-u.ac.jp

**Keywords:** bacteriophage, streptococcus virus, phage therapy, *Streptococcus gordonii*

## Abstract

Bacteriophages are viruses that specifically infect bacteria and are classified as either virulent phages or temperate phages. Despite virulent phages being promising antimicrobial agents due to their bactericidal effects, the implementation of phage therapy depends on the availability of virulent phages against target bacteria. Notably, virulent phages of *Streptococcus gordonii*, which resides in the oral cavity and is an opportunistic pathogen that can cause periodontitis and endocarditis have previously never been found. We thus attempted to isolate virulent phages against *S. gordonii*. In the present study, we report for the first time a virulent bacteriophage against *S. gordonii*, ΦSG005, discovered from drainage water. ΦSG005 is composed of a short, non-contractile tail and a long head, revealing *Podoviridae* characteristics via electron microscopic analysis. In turbidity reduction assays, ΦSG005 showed efficient bactericidal effects on *S. gordonii.* Whole-genome sequencing showed that the virus has a DNA genome of 16,127 bp with 21 coding sequences. We identified no prophage-related elements such as integrase in the ΦSG005 genome, demonstrating that the virus is a virulent phage. Phylogenetic analysis indicated that ΦSG005 forms a distinct clade among the streptococcus viruses and is positioned next to streptococcus virus C1. Molecular characterization revealed the presence of an anti-CRISPR (Acr) IIA5-like protein in the ΦSG005 genome. These findings facilitate our understanding of streptococcus viruses and advance the development of phage therapy against *S. gordonii* infection.

## 1. Introduction

Bacteriophages, which are viruses that infect bacterial cells, are ubiquitous in nature. Phage lifecycles are divided into two major categories, lysogenic and lytic [1,2]. In the lysogenic cycle, temperate phages infect host bacterial cells but are usually integrated into the host genome at attachment sites such as *attP*, *attR*, and *attL* by phage-encoded integrase, which does not lead to bacterial lysis [3]. This integrated form of phage is called a prophage. Conversely, in the lytic cycle, phages never form prophages and can kill host bacterial cells efficiently. Thus, virulent phages, which proliferate remarkably through the lytic cycle, have received significant attention as possible therapeutic agents against bacterial infection. Notably, phages are promising as an anti-microbial resistance (AMR) control option [2,4,5,6], and phages exhibit specific infectivity against bacteria, which allows for phage therapy against bacterial infection without the disruption of associated normal microbiota [7,8,9]. However, whether we can implement phage therapy efficiently depends on the availability of virulent phage. Phage therapy towards some kinds of bacteria has not been developed because virulent phages with efficient bactericidal effects against these bacteria have not been found [10,11,12].

Phage infections are closely associated with host genome diversification. Phages are the most abundant organisms in the biosphere, and prophage sequences have been detected in almost all sequenced bacterial genomes, which indicates that phages drive bacterial evolution via gene transfer and phage resistance. The CRISPR/Cas9 system, which is a gene editing technology platform, was first discovered in *Streptococcus pyogenes* as an adaptive immune response against phage infection [13]. In this system, bacteria can “memorize” exogenous nucleotide sequences and integrate them into the CRISPR array in the form of CRISPR spacers [14]. In addition, prophages or prophage remnants are frequently found in various streptococci genomes [15,16,17]. This viral information derived from the sequenced hosts contributes to detailed understanding of streptococcus viruses; however, there is limited information about virulent phages that infect streptococci compared to other virulent phages such as *Escherichia* viruses and *Pseudomonas* viruses. Notably, virulent phages against Group B and some viridans streptococci such as *S. agalactiae* and *S. gordonii* have not been reported previously.

*S. gordonii* is a gram-positive facultative anaerobe that inhabits the oral cavities of several animal species, including humans [18]. Although *S. gordonii* is categorized as an opportunistic pathogen, it has been reported that this bacterium elicits periodontitis through an initial colonization of the tooth surface, which promotes the aggregation of other oral bacteria, such as *Porphyromonas gingivalis*, leading to pathogenic conditions [19,20]. In addition, oral bleeding allows *S. gordonii* to enter into blood circulation, providing it with the opportunity to reach various organs [18,21]. Previous microbiome analysis has detected *S. gordonii* colonization in the heart valves of infective endocarditis patients [22]. *S. gordonii* is thus a candidate target of the oral microbiota for the control of inflammation.

In the present study, we discovered for the first time a novel virulent phage against *S. gordonii*, ΦSG005, and characterized its molecular features. These findings facilitate our understanding of streptococcus viruses and provide insight into the development of phage therapy against *S. gordonii*.

## 2. Materials and Methods

### 2.1. Bacterial Strains

*S. gordonii* ATCC10558 [23] was grown at 37 °C anaerobically with AnaeroPack (Mitsubishi Gas Chemical, Tokyo, Japan) in Gifu anaerobic medium (GAM) broth (Nissui, Tokyo, Japan) and used for downstream assays as described below.

### 2.2. Bacteriophage Isolation

Phages were isolated by the double-layer method as described elsewhere [4]. In brief, drainage water collected from Kagoshima University School of Dentistry (approved by the Committee on Ethics of Kagoshima University: permit number 180294) was centrifuged at 10,000× *g* for 15 min. The resultant supernatants were filtered through 0.45-µm syringe filters (Advantec, Tokyo, Japan) and subject to polyethylene glycol 6000-NaCl centrifugation. This concentrated sample was mixed with *S. gordonii* ATCC10558 and overlaid on brain heart infusion (BHI) agar (Becton Dickinson, Franklin Lakes, NJ, USA) plates with GAM top agar containing 0.5% agarose ME (Iwai Chemicals, Tokyo, Japan) and incubated at 37 °C anaerobically overnight. Single plaques were collected and purified by repeatedly plating and picking four times.

### 2.3. Bacteriophage Preparation

For downstream assays, an isolated phage was amplified by the double-layer method [4,5]. In brief, *S. gordonii* ATCC10558 in GAM broth was mixed with phage and added to 3 mL of 0.5% GAM top agar. Subsequently, the mixture was overlaid on a BHI agar plate. After overnight incubation of the plate at 37 °C anaerobically, 3 mL of SM buffer (10 mM MgSO_4_, 100 mM NaCl, 0.01% gelatin, and 50 mM Tris-HCl [pH 7.5]) was added to the plate, and the plate was incubated at room temperature for 1 to 2 h with shaking. The overlaid top agar was scraped off and homogenized with SM buffer by a glass spreader. The collected homogenate was purified using an Amicon Ultra membrane filter (Merck, Darmstadt, Germany) based on the phage on tap (PoT) method described by Bonilla et al. [24]. The viral titer of the phage was calculated as the number of plaques in a plaque assay using *S. gordonii* ATCC10558 as the host in accordance with previous reports [4] and is represented as plaque-forming units per milliliter (pfu/mL).

### 2.4. Transmission Electron Microscopic Imaging

Electron microscopic imaging was performed as described previously [5]. Purified phage samples were loaded onto copper grids (EMJapan, Tokyo, Japan). The grids were washed with SM buffer twice and stained with 2% uranyl acetate. Stained samples were observed with a Hitachi HT7700 transmission electron microscope (Hitachi, Ltd., Tokyo, Japan) at 75 kV.

### 2.5. Whole-Genome Sequencing

Purified phage samples were treated with DNase by using a TURBO DNase free kit (Thermo Fisher Scientific, San Jose, CA, USA) following the manufacturer’s protocol. Then, the phage genome was extracted with the Phage DNA isolation kit (Novagen, Nottingham, UK). Libraries were prepared using a Nextere XT DNA library Preparation Kit (Illumina, San Diego, CA, USA). The whole genome of isolated phage was then 300-bp paired-end sequenced on a MiSeq platform (Illumina, San Diego, CA, USA) according to the manufacturer’s instruction. The resulting 324,906 reads were trimmed of adaptors and low-quality bases (end and beginning of reads: Q score < 20, others: Q score < 15), and short reads (<80 bp) were removed using Trimmomatic v0.39, resulting in a total of 300,169 reads (5584 × coverage). The generated sequence reads were assembled by de novo assembly using Unicycler v0.4.8. beta. Finally, the assembled sequences were annotated using PATRIC (https://www.patricbrc.org (accessed on 10 September 2021)) with standard settings. The obtained genomic DNA sequence was submitted to the DDBJ/EMBL/GenBank databases under accession number LC628082 as summarized in Table 1.

### 2.6. Adsorption Assay

Adsorption rates of the phage towards host strains were determined as described in a previous report [5]. Phage in SM buffer (1.0 × 10^8^ pfu/mL) was suspended 1:1 with *S. gordonii* ATCC10558 (1.0 × 10^9^ cfu/mL) at a multiplicity of infection (MOI) of 0.1. After 0.5, 1, 5, or 10 min incubation at room temperature, the samples were immediately centrifuged at 10,000× *g* for 10 min, and the resultant supernatant containing unabsorbed phages was used for the plaque assay described above.

### 2.7. Co-Culture of the Phage and Host Strain

The lytic activity of the phage against *S. gordonii* ATCC10558 was evaluated by turbidity reduction assays as previously reported [4,6]. In brief, phage was inoculated to *S. gordonii* ATCC10558 in mid-exponential phase at an MOI of 10, 1, and 0.1, and the mixture was subsequently incubated at 37 °C for 24 h. The density of the culture was monitored using a plate reader (Sunrise Rainbow Thermos RC, TECAN, Grödig Austria) at OD_590_ every 1 h. After 24 h incubation, the culture was harvested and washed with PBS three times. Thereafter, obtained samples were further incubated at 37 °C anaerobically for 24 h on BHI agar to calculate viable cells as the number of colonies represented as colony-forming units per milliliter (cfu/mL). In addition, in accordance with previous reports [4,25], 3 µL of diluted phage samples (1.0 × 10^9^ to 1.0 × 10^1^ pfu/mL) in SM buffer was dropped onto a BHI agar plate overlaid with *S. gordonii* ATCC10558 to observe the lytic activity of phages by plaque formation.

### 2.8. Bioinformatics Analysis

A phylogenetic tree of phage whole-genomes was constructed by VICTOR [26] using the obtained viral nucleotide sequence. Amino acid sequences of DNA polymerase (CDS11), tail fiber (CDS15), CHAP-domain containing lysin (CDS12), and phage lysin (CDS13) were aligned with representative sequences of other known phages available in the National Center for Biotechnology Information (NCBI) database, and phylogenetic trees were constructed by MEGA X [27]. Prophage sequences within the genome of *S. gordonii* ATCC10558 were annotated and identified by Phage Search Tool Enhanced Release (PHASTER, https://phaster.ca (accessed on 10 September 2021)). Annotation analysis was performed by PATRIC (https://www.patricbrc.org (accessed on 10 September 2021)). The intergenomic similarity between phages was calculated by VIRIDIC [28].

### 2.9. Statistical Analysis

Statistical analysis was calculated by Tukey’s test based on one-way ANOVA analysis from four independent experiments, and significance was set at *p* < 0.01.

## 3. Results

### 3.1. Isolation of Streptococcus Virus ΦSG005

To isolate virulent phages against *S. gordonii*, we collected drainage water from the Kagoshima University School of Dentistry and purified it for phage isolation, because *S. gordonii* mainly inhabits the oral cavity [18]. The double-layer method using *S. gordonii* ATCC10558 as a host resulted in clear plaques as shown in Figure 1A, and subsequently one plaque was picked for isolation. The purified clone was subjected to morphological analysis. Electron microscopic imaging identified phage particles revealing a short, non-contractile tail and a relatively long polyhedral head (Figure 1B), which was classified morphologically as belonging to the *Podoviridae* family according to the report by Ackerman [29]. Long tail fibers were also observed in the electron micrograph (Figure 1B, lower panel), and this isolate was designated as ΦSG005.

### 3.2. Lytic Activity of Streptococcus Virus ΦSG005

We next assessed the lytic activity of ΦSG005. As shown in Figure 2A, more than 50% of ΦSG005 adsorbed to host streptococcal strain ATCC10558 within 1 min, and about 90% had adsorbed within 5 min. Co-culture of ΦSG005 and *S. gordonii* ATCC10558 demonstrated that ΦSG005 (MOI of 10 and 1) lysed the host strain efficiently from 3–5 h post-infection (hpi) and continued to suppress the growth of *S. gordonii* ATCC10558 at 24 hpi (Figure 2B). The viable cells in the culture inoculated with ΦSG005 (MOI of 1 and 0.1) at 24 hpi decreased significantly compared with that of control (Mock). In addition, viable cells in the culture (MOI of 10) at 24 hpi were under the limit of detection (Figure 2C). Figure 2D clearly shows that ΦSG005 produced clear spots at high doses (1.0 × 10^9^–1.0 × 10^6^ pfu/mL) and plaques at low doses (1.0 × 10^5^–1.0 × 10^3^ pfu/mL), representing efficient bactericidal activity as “lysis from within”.

### 3.3. Genome Organization of ΦSG005

Extracted DNA from ΦSG005 was whole-genome sequenced on a short-read platform. The obtained reads assembled through Unicycler resulted in a single linear contig with a total length of 16,127 bp, a G+C content of 34.5%, and 21 coding sequences (CDS) as described in Table 1. The resultant full-length nucleotide sequence was subjected to a BLASTn search, which demonstrated that there are no similar viral sequences in the database, indicating that ΦSG005 possesses a previously uncharacterized genome. Predicted functions of each ΦSG005 CDS are annotated by PATRIC and PSI-BLAST analysis is summarized in Table 2, which shows that ΦSG005 possesses phage-associated genes such as an anti-CRISPR (Acr) IIA5-like protein (CDS2), phage capsid and scaffold (CDS5 and 6), phage tail fiber (CDS11), holin (CDS14), DNA polymerase (CDS15), and phage neck (CDS16). In addition, no phage-associated integrases were detected in the viral genome by gene annotation analysis, indicating that ΦSG005 is a virulent phage. In addition, the genome organization of ΦSG005 implied that CDS5-CDS12 encode viral structural proteins, while CDS13 and CDS14 encode the lysis module of ΦSG005 (Figure 3A). Notably, it was predicted that ΦSG005 possesses two lysin-related genes, a CHAP-domain containing lysin (CDS12) and phage endolysin (CDS13) by PATRIC; however, the ΦSG005 genome organization (Figure 3A) indicated that CDS12 is a tail-associated enzyme categorized as a structural component.

As the presence of Acr in the ΦSG005 was indicated, we then identified the CRISPR-associated sequences in *S. gordonii* ATCC10558 by PATRIC annotation, which revealed that *S. gordonii* ATCC10558 harbors a type II CRISPR-Cas system and a CRISPR array containing 19 repeats and 18 spacers (Figure 3B). The spacer-targeting analysis identified homologous sequences including a 7-nt mismatch at the 5′ region against the ΦSG005 genome at CDS11.

### 3.4. Prophage Elements in S. gordonii ATCC10558

To exclude the possibility that ΦSG005 was derived from the host *S. gordonii* genome, we detected prophage elements in the *S. gordonii* ATCC10558 chromosome, which is composed of 2,187,611 bp with a G+C content of 49.52% and 2039 CDSs (Table 1). Although PHASTER analysis predicted three prophage-related elements in the host genome (Regions 1–3, Appendix A), every region revealed incomplete prophage sequences. Each region was composed of hypothetical proteins (HYP) and phage-like protein (PLP), as shown in Appendix A. Only Region 1 exhibited the presence of attachment sites, *attL* and *attR*. These sequences showed no similarity with that of ΦSG005 in terms of the length, predicted CDSs functions, or genomic organization (Figure 3A and Appendix A).

### 3.5. Molecular Characterization of ΦSG005

Phylogenetic analysis was conducted to clarify the relationships between known streptococcus viruses and ΦSG005. As shown in Figure 4, a phylogenetic tree constructed using whole-genome sequences by VICTOR yielded fifteen species, four genus, and two family clusters. ΦSG005 forms a distinct clade among the streptococcus viruses, but shares common evolutionary origins with streptococcus virus C1, indicating that ΦSG005 is a unique species and independent from any other known streptococcus virus. The intergenomic similarity between streptococcus virus C1 and ΦSG005 was 13.11%. Further phylogenetic analysis was performed using DNA polymerase (CDS15), tail fiber (CDS11), endolysin (CDS13), and CHAP-domain containing lysin (CDS12) sequences to determine the molecular characteristics of ΦSG005. The phylogenetic tree constructed using the DNA polymerase sequence revealed that it formed an independent branch located close to streptococcus virus C1 DNA polymerase (Figure 5A). In addition, phylogenetic analysis of the tail fiber demonstrated that the ΦSG005 tail fiber is most closely related to streptococcus virus C1 tail proteins (Figure 5B). A phylogenetic tree constructed using endolysin resulted in a distinct clade among the streptococcal endolysins, but CHAP-domain containing lysin formed an independent branch and shared ancestral roots with phage lysin derived from streptococcus virus C1 (PlyC, Figure 5C,D). These data indicated that ΦSG005 shares the most recent common ancestor with streptococcus virus C1, but forms a distinct clade among the streptococcus viruses.

## 4. Discussion

In the present study, we discovered a novel virulent streptococcus virus, ΦSG005. Molecular characterization of the virus revealed that ΦSG005 possesses an independent and unique sequence compared to previously known phages, so ΦSG005 expands our understanding of the molecular features of *Streptococcus* viruses. ΦSG005 was found to be related to streptococcus virus C1, which was isolated from a sewage plant in Milwaukee in 1925 as a virulent phage [30]. It has taken nearly 100 years to discover a virus that forms a distinct clade next to streptococcus virus C1; however, our findings make us optimistic that undiscovered streptococcus viruses harboring unique molecular characteristics still exist in nature. In addition, spacer targeting analysis has received attention as a novel prediction tool to search for phages [31,32], as CRISPR spacers are derived from previous infection by phage genomic elements; these sequences guarantee the presence of infectious phages towards host strains. Thus, our analysis indicated the presence of ΦSG005-related streptococcus viruses elsewhere, because the CRISPR tracer 1 sequence was not completely consistent with the ΦSG005 tail fiber sequence, which ensures the presence of diverse streptococcus viruses.

The isolation of phages is affected by several factors. Notably, multiple phage-resistance systems are likely involved in the failure of phage isolations [1,33]. In the present study, we found that the host strain *S. gordonii* ATCC10558 contains a CRISPR spacer showing high sequence similarity with the middle region of the ΦSG005 tail fiber sequence, suggesting that *S. gordonii* ATCC10558 has previously been infected with phages similar to ΦSG005. A previous study reported that *S. pyogenes*-derived Cas9 tolerates nucleotide mismatches between the guide RNA and target DNA. Notably, multiple protospacer adjacent motif (PAM)-distal mismatches can be tolerated for Cas9 nucleotide digestion [13,34]. As shown in Figure 3B, a mismatched 7-nt between CRISPR spacer 1 and the ΦSG005 target DNA position in the PAM-distal region (5′ of CRISPR RNA: crRNA) suggests that this crRNA might recognize the genome of ΦSG005 during infection, leading to phage-resistance; however, our bioinformatics analysis estimated that ΦSG005 encodes an AcrIIA5-like protein, which functions as type II Cas9 inhibitor as a counter-defense mechanism [35]. As *S. gordonii* ATCC10558 harbors a type II CRISPR/Cas system, these data suggested that we could isolate novel virulent phages fortuitously due to the ΦSG005-encoded anti-CRISPR system, because the bacterial adaptive immune response is possibly neutralized by a phage-encoded AcrIIA5-like protein. In order to isolate novel undiscovered lytic phages such as *S. agalactiae* phages, we have to develop further sophisticated isolation methods avoiding phage-resistance mechanisms as represented by CRISPR/Cas9. To this end, further understanding of the counter-defense systems encoded by phages is required.

The bactericidal effects of phages can be applied as treatments for bacterial infections, so-called phage therapy. In these therapies, virulent phages should be applied to achieve desirable outcomes, because temperate phages may transfer toxic or pathogenic genes [2,36]. Thus, whether we can implement phage therapy efficiently depends on the availability of virulent phages. In fact, phage therapy against *S. agalactiae* and *Staphylococcus pseudintermedius* infection has still not been developed well, because virulent phages with efficient bactericidal effects against these bacteria have not been found [10,11,12]. In this context, the first isolated virulent phage against *S. gordonii,* ΦSG005, that possesses a bactericidal effect (Figure 2) has the potential to be developed into a phage therapy against *S. gordonii* infection. Notably, phages exhibit specific infectivity against bacteria, which allows for phage therapy without the disruption of associated normal microbiota. Therefore, further characterization of the ΦSG005 lytic capability against diverse oral bacteria, including different *S. gordonii* isolates, would expand the potential of this virulent phage for therapy. In addition, some kinds of phage-derived recombinant proteins, such as endolysins and tail-associated enzymes, by themselves eliminate or reduce bacteria [37,38], suggesting a unique antimicrobial strategy. Notably, despite streptococcus virus C1 infecting only group C streptococcus [39,40], the endolysin PlyC derived from streptococcus virus C1, which is closely related to CHAP-domain containing lysin (CDS13, Figure 5D), has potent lytic activity against group A, C, and E streptococcus strains [41]. Thus, ΦSG005-encoded lysins may contribute to the development of alternative approaches against *S. gordonii*.

In the present study, we isolated for the first time a virulent streptococcus virus, ΦSG005, that infects *S. gordonii*. These findings facilitate our understanding of streptococcus viruses and have potential for developing phage therapy against *S. gordonii* infection.

## Figures and Tables

**Figure 1 viruses-13-01964-f001:**
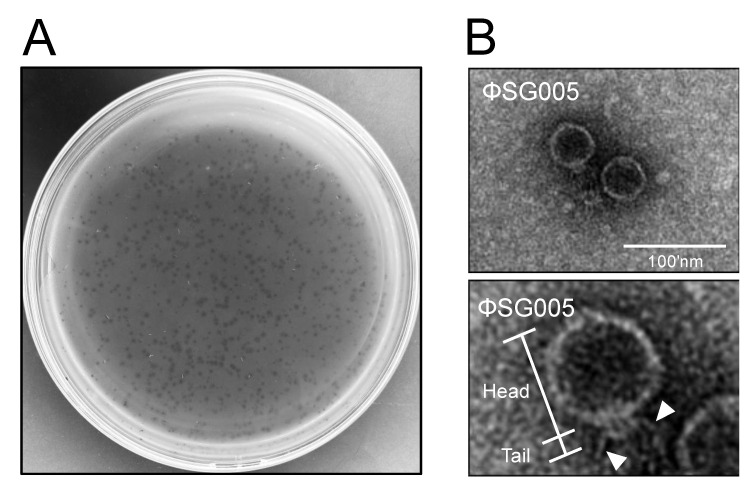
Isolation of a streptococcus virus, ΦSG005. (**A**) Plaques representing phages are observed on a lawn of *S. gordonii* ATCC10558. (**B**) Electron microscopic image showing phage particles exhibiting *Podoviridae* characteristics (upper panel). Lower panel indicates representative morphological structures of the viral particle including the head and tail. Arrowheads indicate the long tail fibers of the virus. Bar = 100 nm.

**Figure 2 viruses-13-01964-f002:**
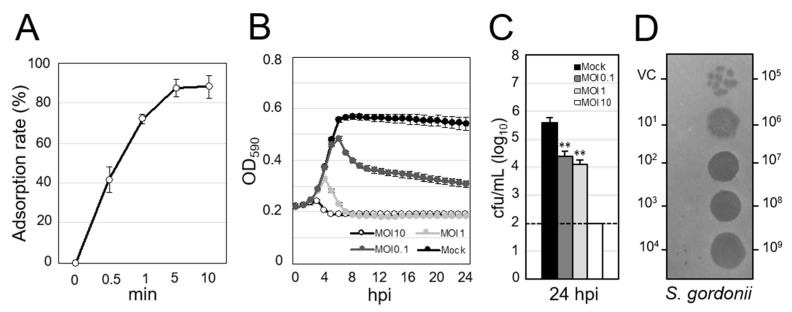
Lytic activity of ΦSG005 against *S. gordonii* ATCC10558. (**A**) Adsorption rate of ΦSG005 on *S. gordonii* ATCC10558 presented as means ± SD (*n* = 4). (**B**) Lytic curves of *S. gordonii* ATCC10558 growing in the presence of ΦSG005 were obtained by monitoring the OD_590_ until 24 hpi. ΦSG005 was inoculated at an MOI of 10, 1, and 0.1. The individual points in each lytic curve are presented as means ± SD (*n* = 4). (**C**) Viable cells from the culture at 24 hpi are presented as means ± SD (*n* = 4). 1.0 × 10^2^ cfu/mL was the limit of detection. Significance was analyzed by Tukey’s test based on one-way ANOVA analysis and is indicated by asterisks (** *p* < 0.01). (**D**) Representative spots and plaques on a lawn of *S. gordonii* ATCC10558 produced by ΦSG005. VC means vehicle control (SM buffer only).

**Figure 3 viruses-13-01964-f003:**
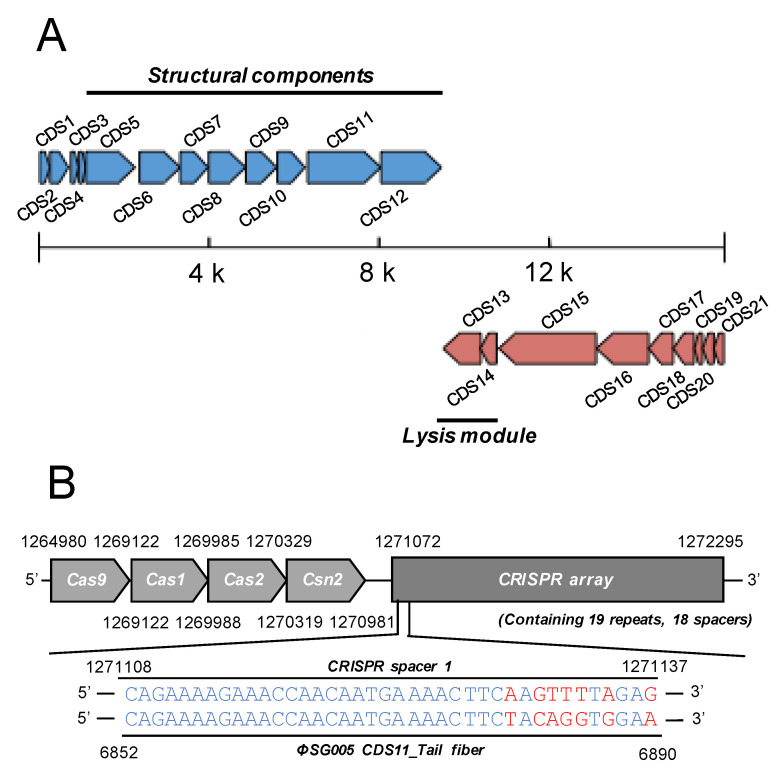
Genome structure of ΦSG005. (**A**) ΦSG005 possesses 16,127 bp and a total of 21 CDSs indicated as blue or red arrows (complementary). Putative structural components (from CDS4 to CDS12) and putative lysis modules (CDS13 and CDS14) are marked by solid lines. (**B**) A section of the CRISPR-associated region in *S. gordonii* ATCC10558 is depicted. *S. gordonii* ATCC10558 harbors the type II CRISPR-Cas system composed of *Cas9, Cas1, Cas2,* and *Csn2* (1,264,980–1,270,981 bp). The CRISPR array contains 19 repeats and 18 spacer sequences. One of the CRISPR spacers located at the most 5′ proximal region in the CRISPR array (represented as CRISPR spacer 1) shows high similarity with the sequence of the ΦSG005 CDS11 predicted tail fiber.

**Figure 4 viruses-13-01964-f004:**
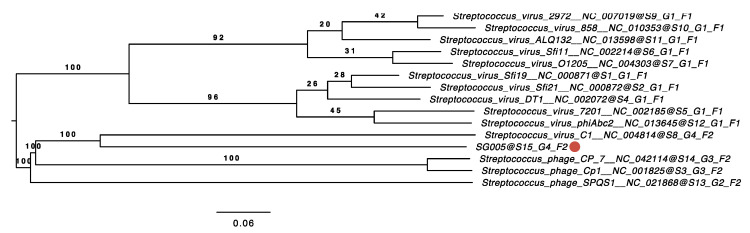
Phylogenetic analysis using the whole-genome sequence of ΦSG005. The phylogenetic tree was constructed by VICTOR using 14 whole-genome sequences of all streptococcus master species registered by the International Committee on the Taxonomy of Viruses (ICTV). The red circle indicates ΦSG005. S, G, and F after the phage names refer to species, genus, and family clusters, respectively.

**Figure 5 viruses-13-01964-f005:**
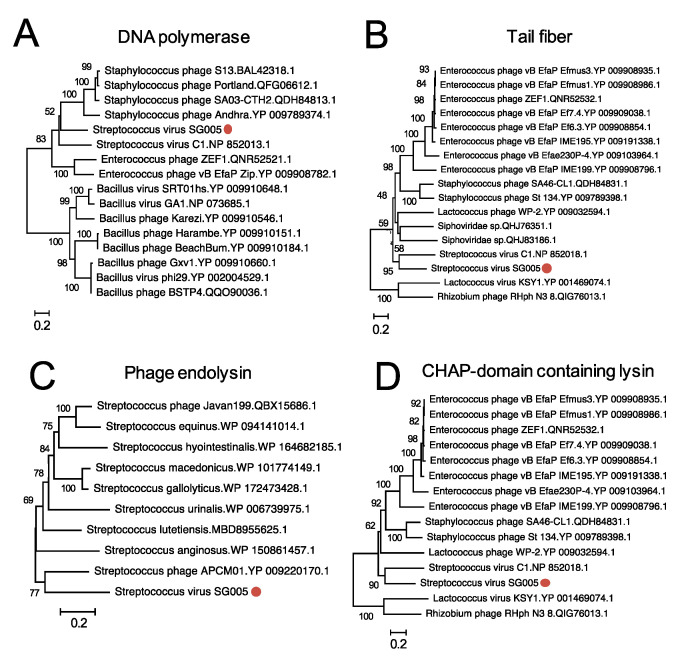
Phylogenetic analysis of some CDSs of ΦSG005. Phylogenetic trees of DNA polymerase (**A**), tail fiber protein (**B**), endolysin (**C**), and CHAP-domain containing lysin (**D**) were constructed by the neighbor-joining method (1000 replications) based on the full-length amino acid sequences. Bootstrap values are shown on the branch nodes, and the scale bar represents a distance of 0.2 substitutions per site. Red circles indicate ΦSG005.

**Table 1 viruses-13-01964-t001:** General genomic information of ΦSG005 and *S. gordonii* ATCC10558. The reference genome of *S. gordonii* ATCC10558 was obtained from the DDBJ/EMBL/GenBank databases.

Strain	Component	Length (bp)	G+C (%)	CDS	Accession No.
ΦSG005	Viral genome	16,127	34.5	21	LC_628082
ATCC10558	Chromosome	2,187,611	49.52	2039	NZ_LS483341.1

**Table 2 viruses-13-01964-t002:** Products and predicted functions of proteins encoded by ΦSG005 CDSs.

CDS	Location (Nucleotides)	Strand	Size (No. of Amino Acids)	Product or Predicted Function
1	1–237	+	79	hypothetical protein
2	239–670	+	144	anti-CRISPR protein AcrIIA5
3	730–906	+	59	hypothetical protein
4	921–1079	+	53	Phage protein
5	1079–2239	+	387	Phage capsid and scaffold
6	2344–3306	+	321	Phage capsid and scaffold
7	3299–3955	+	219	lower collar protein
8	3972–4841	+	290	hypothetical protein
9	4844–5569	+	242	hypothetical protein
10	5687–6234	+	216	hypothetical protein
11	6297–8021	+	575	Phage tail fiber
12	8023–9450	+	476	CHAP domain-containing phage lysin
13	10,375–9485	-	297	Phage endolysin
14	10,745–10,362	-	128	holin protein
15	13,088–10,797	-	764	DNA polymerase (EC 2.7.7.7), phage-associated
16	14,318–13,092	-	409	Phage neck
17	14,900–14,322	-	193	hypothetical protein
18	15,394–14,903	-	164	hypothetical protein
19	15,591–15,400	-	64	hypothetical protein
20	15,853–15,593	-	87	hypothetical protein
21	16,088–15,855	-	78	hypothetical protein

## Data Availability

In this section, please provide details regarding where data supporting reported results can be found, including links to publicly archived datasets analyzed or generated during the study. Please refer to suggested Data Availability Statements in section “MDPI Research Data Policies” at https://www.mdpi.com/ethics (accessed on 10 September 2021). You might choose to exclude this statement if the study did not report any data.

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
