# Peer review of "Novel Virulent Bacteriophage ΦSG005, Which Infects Streptococcus gordonii, Forms a Distinct Clade among Streptococcus Viruses"

_viruses, 2021, doi:10.3390/v13101964_

Round 1
Reviewer 1 Report
The manuscript viruses-1352664 is an article that presents molecular characterization of ØSG005 phage active against Streptococcus gordonii with the potential in the phage therapy especially in the case of infections caused by this anaerobic pathogen.
Comments for authors:
- Line 48-49: I suggest rephrase “always kill host bacterial cells”
- Line 76: Please rephrase “other kinds of bacteria”
- Line 88: What is GAM broth composition?
- Line 101: I suggest change “propagate” to “amplify”.
- Line 101: Is the “plate lysate method” appropriate name of described method?
- Table must be placed after the paragraph where it was first mentioned. I suggest to move the Table 1.
- Line 174: Discover or describe? This sentence needs rephrase.
- Did you need the ethics committee agreement for collecting samples of drainage water?
- Line 331: “and not temperate phages” please rephrase
- The text needs correction by English native speaker.
Author Response
Responses to the comments from the reviewers:
Reviewer 1
General Comments:
The manuscript viruses-1352664 is an article that presents molecular characterization of ØSG005 phage active against Streptococcus gordonii with the potential in the phage therapy especially in the case of infections caused by this anaerobic pathogen.
Response:
We are most grateful to the reviewers for helpful comments, which have allowed us to extend our study. We have addressed the reviewers’ comments on a point-by-point basis, we improved our manuscript and added explanation. Please confirm our response described below. Modified sentences were highlighted in yellow color in the revised text.
Detailed Comments:
- Line 48-49: I suggest rephrase “always kill host bacterial cells”
Response:
Thank you so much for the comment. As the reviewer mentioned that, the phrase “always kill” is inappropriate. We replaced “always kill host bacterial cells” with “can kill host bacterial cells efficiently” in the present manuscript, L50, page 2.
- Line 76: Please rephrase “other kinds of bacteria”
Response:
Thank you for the comment. We replaced “other kinds of bacteria” with “other oral bacteria, such as Porphyromonas gingivalis” in the present manuscript, L78, page 2.
- Line 88: What is GAM broth composition?
Response:
Thank you so much for kind confirmation. We spelled out “GAM” (Gifu anaerobic medium) in present the manuscript, L91, page 2. In addition, composition is described below.
Ingredients (Gms / L) : Peptic digest of animal tissue (10.0), Papaic digest of soyabean meal (3.0), Proteose peptone (10.0), Digested serum (13.5), Yeast extract (5.0), Beef extract (2.2), Liver extract (1.2), Dextrose (3.0), Potassium dihydrogen phosphate (2.5), Sodium chloride (3.0), Starch, Soluble (5.0), L-Cysteine hydrochloride (0.3), Sodium thioglycollate (0.3) Final pH: 7.1
- Line 101: I suggest change “propagate” to “amplify”.
Response:
Thank you for the comment. We replaced “propagate” with “amplified” in the present manuscript, L105, page 3.
- Line 101: Is the “plate lysate method” appropriate name of described method?
Response:
Thank you for the comment. “Double-layer method” is right. We replaced “plate lysate method” with “double-layer method” in the present the manuscript, L105, page 3.
- Table must be placed after the paragraph where it was first mentioned. I suggest to move the Table 1.
Response:
Thank you so much for kind confirmation. Table 1 was moved to the paragraph where it was first mentioned (2.5. Whole genome sequencing) in the present manuscript, page 3.
- Line 174: Discover or describe? This sentence needs rephrase.
Response:
Thank you for the reviewer’s suggestion. We modified the sentences according to the reviewer’s comment, as follows: “To isolate virulent phages against S. gordonii, we collected drainage water from the Kagoshima University School of Dentistry and purified it for phage isolation, because S. gordonii mainly inhabits the oral cavity.” in the present the manuscript, L181-L183, page 4.
- Did you need the ethics committee agreement for collecting samples of drainage water?
Response:
Thank you for the kind confirmation. We added accession no for collecting samples of drainage water (180294) in the present the manuscript, L95-L96, page 2.
- Line 331: “and not temperate phages” please rephrase
Response:
Following the reviewer’s comment, we modified the sentences as follows: “virulent phages should be applied to achieve desirable outcomes, in the present the manuscript, L347-L348, page 10.
- The text needs correction by English native speaker.
Response:
Thank you so much for the comment. As the reviewer pointed out, our present manuscript was reviewed by an English native speaker. In addition, we attached the native check letter.
Reviewer 2 Report
In this paper, the authors present the first characterization of a lytic (virulent) phage of Streptococcus gordonii. The authors characterize the phage genome, the kinetics of infection of S. gordonii strain ATCC10558, and the phylogeny of this phage. Given the overall interest in the evolution of pathogens and the role of bacteriophage in alternative therapies to infection, any attempt to characterize bacteriophage of this opportunistic oral pathogen should be of interest to the readers. The authors are to be commended for a thorough, well-written manuscript; the addition of the insights into the CRISPR system in S. gordonii were a bonus to the interesting description of this new virulent phage. I recommend this paper for publication, while also suggesting that the authors consider some of the points below to enhance the overall clarity and strengthen the relevance of this work.
Minor points
Lines 47-49—temperate phages can lyse the cell, obviously, when reentering the lytic cycle following induction; the term ‘lytic’ is used to describe the second type of replication cycle, but then the word ‘virulent’ is substituted in the actual description and elsewhere in the manuscript—for clarity, please indicate that ‘lytic’ phage are considered virulent (or equate the two words in some other way).
Line 53—‘availability’ might be better than isolation
Table 1—any comment on why the GC content of the phage is so different than the host strain? Is that typical of lytic phage in Streptococcus? Is C1 similarly different in GC composition to its hosts? Does S. gordonii have any known codon (or tRNA availability) bias that might regulate the speed at which these lytic phages can replicate? (This is not an important point for the suitability of the manuscript, but I was struck by the large difference in GC content.)
Line 178—how many plaques were examined by EM and were all identical (in other words, is this the only phage that was recovered, or just the only one that met the criteria for further analysis)?
Figure 2D—Please revisit to consider alternative organizations; because the left has so few plaques (or zones of clearing), it took a minute to realize that there were two columns there; might be better organized horizontally? Or flipping the order (VC at top left and 10-9 at bottom right)? If the consensus is that this is the best way, that’s fine.
Line 280—This is mentioned briefly in the discussion, but I was struck that phage C1 is lytic, of a similar size to φSG005, and morphologically similar, yet appears evolutionarily distinct; not only do the similarities support the role of φSG005 as a lytic phage, but they also offer tantalizing insight into the evolutionary history presumably shared between these two phages. May want to highlight these other similarities, as well. Were any directed experiments done with C1 and φSG005 at the same time? As mentioned in the discussion, C1 has specificity for Group C streptococci, but the difference between the two phages in some of your assays might be striking and highlight the specificity and suitability of a phage such as φSG005 for phage therapy?
Most major comment
Line 102—What is the similarity between the ATCC strain used in this study and other S. gordonii strains? You have selected lytic phage on the ATCC strain, is it lytic in other strains? One of the benefits (and limitations) of phage therapy is the exquisite specificity of a virulent phage to its host. Are there other genotypes of S. gordonii? If not, this should be mentioned (to highlight the potential use of this phage in therapy); if so, then it would have been nice to see φSG005 tested against other strains of S. gordonii to assess how broadly active this phage might be.
Author Response
General Comments:
In this paper, the authors present the first characterization of a lytic (virulent) phage of Streptococcus gordonii. The authors characterize the phage genome, the kinetics of infection of S. gordonii strain ATCC10558, and the phylogeny of this phage. Given the overall interest in the evolution of pathogens and the role of bacteriophage in alternative therapies to infection, any attempt to characterize bacteriophage of this opportunistic oral pathogen should be of interest to the readers. The authors are to be commended for a thorough, well-written manuscript; the addition of the insights into the CRISPR system in S. gordonii were a bonus to the interesting description of this new virulent phage. I recommend this paper for publication, while also suggesting that the authors consider some of the points below to enhance the overall clarity and strengthen the relevance of this work.
Response:
We appreciate the reviewer’s valuable suggestions. We have addressed the reviewers’ comments on a point-by-point basis and improved our manuscript and added explanation. Please confirm our response described below. Modified sentences were highlighted in yellow color in the revised text.
Detailed Comments:
- Lines 47-49—temperate phages can lyse the cell, obviously, when reentering the lytic cycle following induction; the term ‘lytic’ is used to describe the second type of replication cycle, but then the word ‘virulent’ is substituted in the actual description and elsewhere in the manuscript—for clarity, please indicate that ‘lytic’ phage are considered virulent (or equate the two words in some other way).
Response:
Thank you so much for the grateful suggestion. As the reviewer mentioned, it is important to describe lytic and lysogenic cycle correctly. Following reviewer’s comment, we modified the sentence as follow: “Conversely, in the lytic cycle, phages never form prophages and can kill host bacterial cells efficiently. Thus, virulent phages, which proliferate remarkably in the lytic cycle, have received significant attention as possible therapeutic agents against bacterial infection.” in the present manuscript, L49-L52, page 2.
- Line 53—‘availability’ might be better than isolation
Response:
Thank you for the suggestion. We completely agree with this. We modified the sentence as follow: “whether we can implement phage therapy efficiently depends on the availability of virulent phage” in the present manuscript, L55-L56, page 2. We modified the sentence in the present manuscript, L26-L27, page 1 as well”
- Table 1—any comment on why the GC content of the phage is so different than the host strain? Is that typical of lytic phage in Streptococcus? Is C1 similarly different in GC composition to its hosts? Does S. gordoniihave any known codon (or tRNA availability) bias that might regulate the speed at which these lytic phages can replicate? (This is not an important point for the suitability of the manuscript, but I was struck by the large difference in GC content.)
Response:
Thank you so much for the interesting comment. Following the reviewer’s comment, we compared with GC content of C1 and SG005. As a result, GC content of Streptococcus virus C1 was 34.6%, which was very similar with SG005 (GC 34.5%, Table 1). We thus speculated that C1-like phages possess lower GC content than that of the host strain; however, we didn’t isolate other C1-like viruses and perform enough comparison. It’s just hypothesis now. So, we did not include the result regarding comparison of GC content between C1 and SG005 in the present manuscript.
- Line 178—how many plaques were examined by EM and were all identical (in other words, is this the only phage that was recovered, or just the only one that met the criteria for further analysis)?
Response:
Thank you for the comment. We just picked one plaque from the plate. In order to describe correctly, we modified sentence as follow: “and subsequently one plaque was picked for isolation. The purified clone was subjected to morphological analysis.” in the present manuscript, L184-L186, page 4.
- Figure 2D—Please revisit to consider alternative organizations; because the left has so few plaques (or zones of clearing), it took a minute to realize that there were two columns there; might be better organized horizontally? Or flipping the order (VC at top left and 10-9 at bottom right)? If the consensus is that this is the best way, that’s fine.
Response:
Thank you so much for the kind comment. Following the reviewer’s comment, we flipped the order (VC at top left and 10^9 at bottom right) in the present Fig. 2D.
- Line 280—This is mentioned briefly in the discussion, but I was struck that phage C1 is lytic, of a similar size to φSG005, and morphologically similar, yet appears evolutionarily distinct; not only do the similarities support the role of φSG005 as a lytic phage, but they also offer tantalizing insight into the evolutionary history presumably shared between these two phages. May want to highlight these other similarities, as well. Were any directed experiments done with C1 and φSG005 at the same time? As mentioned in the discussion, C1 has specificity for Group C streptococci, but the difference between the two phages in some of your assays might be striking and highlight the specificity and suitability of a phage such as φSG005 for phage therapy?
Response:
Thank you so much for the critical point. In the present study, we did not perform experiments using C1 and SG005 at the same time. Then, actually it was difficult to compare with host specificity or lytic activity between C1 and SG005. On the other hand, as the reviewer mentioned, host range of SG005, whether it is limited in S. gordonii or not, is highly interesting; however, S. gordonii belongs to the viridans streptococci, which is different from Group C streptococci, indicating that lytic capability of C1 and SG005 might have differences. For instance, it has been reported that C1 derived endolysin (PlyC) reveals specific efficient lytic activity against Group A, C and E streptococci, whereas it results in little or no lytic activity against oral streptococci such as S. gordonii (Ref No. 41., PMID: 11259652), suggesting that C1 and SG005 might have distinct lytic features. Therefore, next, we initially have to investigate the lytic capability of SG005 against oral streptococci including various S. gordonii clinical isolates (we’re now collecting) and other group streptococci, which would contribute to widen the therapeutic potentials of SG005. Related to this comment and the other major comment from the reviewer #2 (described below), we further discussed about this in the present manuscript, L355-L359, page 10.
- Line 102—What is the similarity between the ATCC strain used in this study and other S. gordonii strains? You have selected lytic phage on the ATCC strain, is it lytic in other strains? One of the benefits (and limitations) of phage therapy is the exquisite specificity of a virulent phage to its host. Are there other genotypes of S. gordonii? If not, this should be mentioned (to highlight the potential use of this phage in therapy); if so, then it would have been nice to see φSG005 tested against other strains of S. gordoniito assess how broadly active this phage might be.
Response:
Thank you for the suggestion regarding a very critical point. As we described above, we used one S. gordonii strain (ATCC10558 assigned as a host for phage isolation) and did not compare lytic activity of SG005 with other phages, also did not determine SG005 host specificity. We completely agree with the reviewer’s comment “One of the benefits (and limitations) of phage therapy is the exquisite specificity of a virulent phage to its host”. We thus mentioned this point and further discussed in the present manuscript, L355-L359, page 10.
Reviewer 3 Report
Work from Fujiki et al. describes the isolation of the first lytic phage against S. gordonii. Although study has impact in the area of bacteriophage therapy against opportunistic pathogens, there are some aspects that need to be addressed and/or clarified:
In general terms text is difficult to follow, especially introduction and discussion. Although results and methodology are presented more clearly the whole text need to be extensively revised as it is complicated to understand the message as it is confusing and ideas not well presented.
Abstract is also confusing and text does not flow.
It is unclear, under my understanding, what the taxonomic classification of the phage isolated is. Although it is mentioned that morphologically it is a podovirus, it should be clear, when doing WGS, what is the actual classification (Kraken or other tool for taxonomic ID based on reads has been done?).
Fig 5., probably not needed as there is a tree for the whole genome and it does not add anything new to the results (or I cannot see the exact value of this figure, please clarify).
Author Response
General Comments:
Work from Fujiki et al. describes the isolation of the first lytic phage against S. gordonii. Although study has impact in the area of bacteriophage therapy against opportunistic pathogens, there are some aspects that need to be addressed and/or clarified:
Response:
We appreciate the valuable reviewer’s comments on our manuscript. As the reviewer pointed out, we improved our manuscript and added explanation. Please confirm our response described below. In addition, following the reviewer’s comments, we performed editing of English language and style of the present manuscript. We attached native check letter. Modified sentences were highlighted in yellow color in the revised text.
Detailed Comments:
- In general terms text is difficult to follow, especially introduction and discussion. Although results and methodology are presented more clearly the whole text need to be extensively revised as it is complicated to understand the message as it is confusing and ideas not well presented. Abstract is also confusing and text does not flow.
Response:
Thank you so much for the kind comment. Following the reviewer’s comment, our present manuscript was reviewed by a native English speaker. In addition, native check letter is attached. Notably, objective of our study is characterization of a novel virulent phage against S. gordonii. Actually, given that virulent phages against S. gordonii were not found and reported previously, we believe that molecular characterization of ΦSG005 will help to facilitate a greater understandings of streptococcus viruses. In addition, lytic phages have been received significant attentions for therapy against bacterial infection owing to major two reasons. First, phages possess promising potency as anti-microbial resistance (AMR) control options. Generally, second, phages exhibit specific infectivity against bacteria, which allow us to perform phage therapy in bacterial infection without disruption of associated normal microbiota. In this context, our findings provide insight into the development of phage therapy against S. gordonii. Regarding second point, we further discussed about this and added explanation in the present manuscript, L355-L359, page 10.
- It is unclear, under my understanding, what the taxonomic classification of the phage isolated is. Although it is mentioned that morphologically it is a podovirus, it should be clear, when doing WGS, what is the actual classification (Kraken or other tool for taxonomic ID based on reads has been done?). 3. Fig 5., probably not needed as there is a tree for the whole genome and it does not add anything new to the results (or I cannot see the exact value of this figure, please clarify).
Response (2 and 3):
Thank you for the comments. The International Committee on Taxonomy of Viruses (ICTV), which established rules for the naming and classification of viruses, published master species list of prokaryotic viruses assigned to 548 species, 103 genera, 7 subfamilies and 18 families in 2015. VICTOR (performed in Fig.5) is genome-based phylogeny and classification of prokaryotic viruses and perform taxonomic classification (Ref. No. 26., PMID: 29036289). In the present study, we performed viral classification by VICTOR using 14 whole-genome sequences of all streptococcus master species registered by the ICTV. Phylogenetic analysis by VICTOR demonstrated that ØSG005 forms a distinct clade among the streptococcus viruses, but shares common evolutionally origins with streptococcus virus C1 and yielded fifteen species, four
genus, and two family clusters (S_G_F after the phage name refers to species, genus, and family clusters, respectively). In order to clarify the meaning of Fig.5, we added detailed explanations in the present manuscript, L276-L278 and L300-L301, page 10. This method was used for phage classification previously (Horvath M et al. 2020. Sci Rep. PMID: 32246126., Fujiki J et al. 2020. Microbiol Immunol. PMID: 32918505., Salem M et al. 2018. Viruses. PMID: 29614052. etc).
Reviewer 4 Report
I think that the identification of an anti-CRISPR protein in this paper is an important example. The publication of this article is important for the bacteriophage study.
The authors identified a new phage for S. gordonii , ØSG005, and analyzed its genome. As a result of genome analysis, they found that the genome of ØSG005 formed an independent and unique sequence and that host S. gordonii contained a CRISPR spacer with high sequence similarity with ØSG005 tail fiber. These results are of very interest in studying the relationship between phages and hosts. So, I recommend this manuscript for publication.
Author Response
General Comments:
I think that the identification of an anti-CRISPR protein in this paper is an important example. The publication of this article is important for the bacteriophage study. The authors identified a new phage for S. gordonii, ØSG005, and analyzed its genome. As a result of genome analysis, they found that the genome of ØSG005 formed an independent and unique sequence and that host S. gordonii contained a CRISPR spacer with high sequence similarity with ØSG005 tail fiber. These results are of very interest in studying the relationship between phages and hosts. So, I recommend this manuscript for publication.
Response:
Thank you so much for the reviewer’s comments on our manuscript. Notably, we appreciate the valuable comment for anti-CRISPR protein.